# Combined Widely Targeted Metabolomic, Transcriptomic, and Spatial Metabolomic Analysis Reveals the Potential Mechanism of Coloration and Fruit Quality Formation in *Actinidia chinensis* cv. Hongyang

**DOI:** 10.3390/foods13020233

**Published:** 2024-01-11

**Authors:** Jipeng Mao, Zhu Gao, Xiaoling Wang, Mengfei Lin, Lu Chen, Xinyi Ning

**Affiliations:** 1Jiangxi Kiwifruit Engineering Research Center, Institute of Biological Resources, Jiangxi Academy of Sciences, Nanchang 330096, China; maojipeng@jxas.ac.cn (J.M.);; 2Jinggangshan Institute of Biotechnology, Jiangxi Academy of Sciences, Ji’an 343009, China; chenlu@jxas.ac.cn; 3College of Environmental and Chemical Engineering, Nanchang Hangkong University, Nanchang 330063, China

**Keywords:** *Actinidia chinensis* cv. Hongyang, fruit color, fruit quality, metabolome, transcriptome

## Abstract

Postharvest kiwifruit (*Actinidia chinensis* cv. Hongyang) pulp is mainly composed of outer yellow-flesh (LR) and inner red-flesh (HR). However, information about the differences in coloration and fruit quality between these two parts are limited. In this study, widely targeted metabolomic, transcriptomic, and spatial metabolomic analyses were used to reveal the potential mechanism of coloration and fruit quality formation. The results show that a total of 1001 metabolites were identified in Hongyang kiwifruit, and the accumulation of 211 metabolites were significantly higher in the HR than LR, including 69 flavonoids, 53 phenolic acids, and 38 terpenoids. There were no significant differences in the content of citric acid, quinic acid, glucose, fructose, or sucrose between the LR and HR. These results were consistent with the results from the RNA-seq profile and spatial metabolomic analysis. In addition, a total of 23 key candidate genes related to flesh color and fruit quality formation were identified and validated by qRT-PCR analysis. This study provides a theoretical basis for elucidating the underlying mechanism of the formation of kiwifruit flesh color and fruit quality.

## 1. Introduction

Kiwifruit (*Actinidia* spp.) is one of the most successfully cultivated and domesticated fruit species in the 20th century. Kiwifruit is rich in amino acids, flavonoids, phenolic acids, and vitamins, especially vitamin C [1,2]. A total of 54 species and 21 varieties in the genus of *Actinidia* have been recognized worldwide, in which 52 species are native to China [3]. Based on the rich kiwifruit germplasm resources, our county has bred more than 200 varieties of kiwifruit. *Actinidia chinensis* cv. Hongyang, the world’s first red-flesh kiwifruit variety bred in our country, has the characteristics of early maturity, rich nutrition, high soluble solid content (SSC), excellent fresh quality, and is highly appreciated by consumers. With the development of metabolomics technology, metabolite components of more than 20 cultivars of kiwifruit, including Hongyang have been identified in recent years. In particular, 515 metabolites were identified in Hongyang kiwifruit [4], 285 metabolites were identified in Jinshi 1 kiwifruit [2], and 48 volatile compounds were identified in Hort 16A, Jinyan, and Guichang kiwifruit [5]. In addition, comprehensive metabolic map and genome-wide transcriptome analysis revealed the major quality regulations and regulatory network involved in flavor formation of Hongyang kiwifruit [4]. However, so far little is known about the spatial distribution and variation of key flavor components in the postharvest fruit of kiwifruit.

In addition to being a fresh fruit, kiwifruit is also an excellent raw material for the production of functional health foods due to its high content of functional components, such as flavonoids, phenolic acids, and terpenoids [6]. As one of the important active components of kiwifruit, flavonoids have functions of anti-oxidation, anti-viral, liver protection, anti-inflammatory, and prevention of cardiovascular diseases [7]. In addition, there are several flavonoid compounds involved in the formation of fruit flavor and color, such as 4β-(S-cysteinyl)-epicatechin/catechin, naringenin chalcone, esculeoside A/B for flavor formation [8], and cyanidin 3-O-glucoside, pelargonidin 3-O-glucoside, quercetin rutinoside, kaempferol 3-O-glucoside for color formation [9,10,11]. Phenolic acid are non-flavonoid polyphenols, which are widely found in plants. There are often combined with other polyphenols and polysaccharides, and a small amount exists in free form. Phenolic acids have many positive effects on human health, such as anti-oxidation, anti-inflammatory, anti-bacterial, anti-viral, and anti-diabetic activities [12]. In addition, phenolic acids, such as coumaric acid, coutaric acid, vanillin, and syringic acid, have an effect on fruit quality [13]. Terpenoids are the largest class of natural compounds and are widely used in the food and pharmaceutical industries. They have a variety of biological activities, such as anti-fungal, anti-malarial, anti-viral, and anti-inflammatory activities [12]. Volatile terpenoids such as limonene, β-cyclocitral, linalool, benzaldehyde, and pulegone contribute to the aroma of the fruits [14].

Metabolomics is an important component of systems biology that can be used to identify and quantify endogenous small molecule metabolites [15]. It is often combined with transcriptomics for candidate gene screening, elucidation of metabolite biosynthetic pathways, and construction of metabolic regulatory networks [16,17]. However, untargeted or targeted metabolomics techniques based on UPLC-ESI-MS/MS [18] or GC-MS [19,20] can only identify and quantify metabolites, without the function of visualizing target metabolites. Spatial metabolomics technology, relying on matrix-assisted laser desorption/ionization mass spectrometry imaging (MALDI-MSI), is the most advanced tissue imaging technology, which can visualize the spatial distribution of metabolite components in different tissues without complex sample pre-processing procedures and reference labels [21,22,23]. This can break the limitations imposed by conventional methods that require pulverization, extraction, and purification [24,25]. Recently, the MALDI-MSI technique has been extensively used to analyze the accurate spatial distribution and dynamic changes of metabolites [26,27], polyphenols [28,29], esters [30], lipids [31,32,33], glycoalkaloids [34,35,36], terpenoids [37], flavonoids [25,38,39,40,41], amino acids [38,40], sugars [39,40,42], and organic acids [38,39] in a variety of Chinese medical herbs, crops, and horticultural plants. However, so far, no application of spatial metabolomics techniques with the spatial distribution of kiwifruit metabolites has been reported.

Flesh color is an important characteristic of fruit quality, thus revealing the formation mechanism of flesh color and its influence on fruit quality has important theoretical significance and practical application value. *Actinidia chinesis* cv. Hongyang is the main backbone parent of kiwifruit breeding. In this study, UPLC-ESI-MS/MS and transcriptome sequencing (RNA-seq) techniques were used to analyze the differences in metabolite components and content and the expression levels of related genes in different tissues of Hongyang kiwifruit. In addition, we established a highly sensitive MALDI-MSI method suitable for kiwifruit tissues to visualize the target metabolites. The spatial distribution of key flavor compounds in Hongyang kiwifruit and the dynamic changes during the postharvest softening process were visualized and analyzed. Our results provide a theoretical basis for elucidating the underlying mechanism of the formation of kiwifruit flesh color and fruit quality.

## 2. Materials and Methods

### 2.1. Plant Materials

*Actinidia chinensis* cv. Hongyang fruit samples that reached physiological maturity (140 d after pollination, SSC greater than 6.5) were obtained from Fengxin County Doctor Kiwifruit Base (E 114°45, N 28°34′), Yichun City, Jiangxi Province, China. Collected Hongyang kiwifruit were immediately transported to the Laboratory of Kiwifruit Engineering Research Center of Jiangxi province. After dissipating the field heat, they were stored at 20 °C, and sampled on d 0 (harvested stage, phase I), d 6 (rapid softening stage, phase II), and d 12 (edible stage, phase III). The four sub-tissues; peel (about 1 mm thick) (GP), yellow-flesh (LR), red-flesh (without seed) (HR), and core (GX), from three ripening stages were uniformly mixed and rapidly frozen in liquid nitrogen for metabolite and total RNA extraction (Figure 1A). Sliced tissue from three ripening stages were used for MALDI-MSI analyses.

### 2.2. Total RNA Extraction and Transcriptome Sequencing Analysis

Total RNA from each sample was extracted using a Plant RNA Purification Reagent (Invitrogen, Carlsbad, CA, USA) according to the manufacturer’s instructions. The cDNA libraries were constructed according to the protocol reported by Li et al. [43], and quality checks were performed using Qubit 2.0 software and Q-PCR methods. Libraries were sequenced using the Illumina Novaseq 6000 platform at the Science Corporation of Gene (Guangzhou, China). After filtering out low-quality reads, the clean reads were aligned and assembled using HISAT2.2.1 software [44] and the Trinity method [45], respectively. Based on sequence similarity, the assembled genes were compared against the Nr, KEGG, Swiss-Prot, KOG, and GO public databases to obtain functional annotation information. Identification of differentially expressed genes (DEGs) and weighted gene co-expression network analysis (WGCNA) were performed using the R package DESeq 1.38.0 and WGCNA, respectively. A total of 12 genes were randomly selected for expression analyses by quantitative real-time PCR (qRT-PCR). The primer sequence details for the 12 genes and the internal control gene *AcActin* are shown in Appendix A.

### 2.3. Metabolite Extraction and Widely Targeted Metabolome Analysis

Metabolites were extracted according to the methods described previously, with some modifications [17,18]. The samples were freeze-dried and crushed using a mixer mill (MM400, Retsch, Haan, Germany) with zirconia beads for 1.5 min at 30 Hz. The powder (100 mg) was weighed and submerged in 1.2 mL of 70% aqueous methanol, vortexed for 30 s every 30 min for 6 times in total, and the sample was stored in a refrigerator at 4 °C overnight. Following centrifugation at 12,000 rpm for 10 min, the supernatant was absorbed and filtered (0.22-µm pore size) for UPLC-ESI-MS/MS (UPLC: Shimadzu, Tokyo, Japan; MS: Applied Biosystems, Waltham, USA) analyses by Wuhan Metware Biotechnology Co., Ltd., Wuhan, China.

The sample extracts were analyzed based on the UPLC-ESI-MS/MS system. The analytical conditions were conducted as previously described [17,46,47], using a UPLC column, (Agilent SB-C18, 1.8 µm, 2.1 × 100 mm), with a mobile phase consisting of solvent A, which was pure water with 0.1% formic acid, and solvent B, which was acetonitrile with 0.1% formic acid. The gradient program was 95% A and 5% B at 0 min, a linear gradient to 5% A and 95% B within 9 min, then 5% A and 95% B for 1 min, and 95% A and 5% B adjusted within 0.1 min, and maintained for 2.9 min. The flow rate was 0.35 mL/min with a column temperature of 40 °C, and an injection volume of 4 µL. The MS/MS conditions used an electrospray ionization temperature of 550 °C, ion spray voltages of 5500 V (positive ion mode) and −4500 V (negative ion mode), ion source gas I (50 psi), gas II (60 psi), and curtain gas (25 psi), and the collision-activated dissociation was set to high. Each ion pair was scanned based on the optimized collision energy and declustering potential in the triple quadrupole (QQQ).

Qualitative and quantitative analyses of the metabolites were performed using secondary spectral information based on the public metabolite database and the self-built MWBD database (Wuhan Metware Biotechnology Co., Ltd., China). The characteristic ions of each substance were screened out by the multiple reaction monitoring of QQQ [48], and the signal strengths of the characteristic ions were obtained in the detector. The Analyst 1.6.3 software was used to calculate the relative content of the corresponding substances in the peak area of each chromatographic peak. Multivariate principal component analysis (PCA) and orthogonal partial least squares-discriminant analysis (OPLS-DA) were conducted using the base package and MetaboAnalystR in R. The multivariate analysis of variable importance in projection (VIP) in the OPLS-DA model was used to initially screen differentially accumulated metabolites (DAMs). 

### 2.4. Correlation Analysis of the Transcriptome and Metabolome Data

Pearson’s correlation tests were used to analyze the correlations between the DEGs and DAMs obtained from each comparison group. Correlation results with a Pearson’s correlation coefficient (PCC) value greater than 0.8 and *p*-value less than 0.05 were considered to be significant. Based on the software packages ggplot2 and geropt in R 4.1.0, a nine-quadrant graph was used to demonstrate the consistency of the expression and accumulation patterns of DEGs and DAMs in each comparison group. In addition, the DEGs and DAMs were mapped to the KEGG pathway database to obtain their common pathway information.

### 2.5. Frozen Section Preparation and Mass Spectrometry Imaging

Slice preparation and mass spectrometry parameters were performed according to previously described protocols [49]. Frozen tissue samples were fixed in embedding agent during the cutting stage. Tissues were sectioned at 14 μm thickness using a Leica CM1950 cryostat (Leica Microsystems GmbH, Wetzlar, Germany) at −20 °C. Afterwards, the thawed sections were placed in groups on electrically conductive slides coated with indium tin oxide (ITO), and the slides were dried in a vacuum desiccator for 30 min. MALDI-MSI experiments were performed on a rapifleX MALDI-TOF/TOF MS (Bruker Daltonics, Billerica, MA, USA) equipped with a 10 kHz smartbeam 3D laser. Laser power was set to 90% and then fixed throughout the whole experiment. The mass spectra were acquired in positive reflectron mode, with an accelerating voltage of 20.00 kV, a lens voltage of 11.35 kV, and a reflector voltage of 20.85 kV. The mass spectra data were acquired over a mass range from *m*/*z* 200–600 Da. The imaging spatial resolution was set to 50 μm for the tissue, and each spectrum consisted of 100 laser shots. MALDI mass spectra were normalized with the total ion current (TIC), and the signal intensity in each image was shown as the normalized intensity. MS/MS fragmentations performed on the rapifleX MALDI-TOF/TOF MS in the LIFT mode were used for further detailed structural confirmation of the identified metabolites.

## 3. Results

### 3.1. Metabolomic Profiling

The UPLC-ESI-MS/MS technology platform was used to analyze the differences of metabolite components and content in different sub-tissues of Hongyang kiwifruit. The total Ion Current of quality control samples showed high stability, clear peak pattern, and strong reliability (Appendix A). A total of 1001 metabolites were identified in Hongyang kiwifruit, mainly flavonoids, phenolic acids, lipids, terpenoids, and amino acids and derivatives, with 953,932,937, and 943 metabolites being identified in the GP, LR, HR, and GX, respectively (Appendix A), mainly flavonoids, phenolic acids, lipids, terpenoids, amino acids and derivatives, alkaloids, organic acids, saccharides, and alcohols (Table 1). Among them, 4,7,9,9′-tetrahydroxy-3,3′-dimethoxy-8-O-4′-neolignan and costic acid were specifically accumulated in the HR, while threose, 30-norhederagenin, decanoyl vanillylamide, 5′-methoxyisolariciresinol-9′-O-xyloside, 17 triterpene, 5 coumarins, 3 flavonoids, 3 phenolic acids, and 2 free fatty acids were specifically accumulated in the GP (Appendix A).

PCA was used to elucidate the overall metabolite differences between groups as well as the variability between samples within groups. The results showed that PC1 and PC2 could explain 48.77% and 17.42% of the total variance of the samples, respectively, indicating that there were obvious metabolite differences in different sub-tissues (Figure 1B). The heatmap of hierarchical clustering analysis was used to classify the accumulation and variation model of metabolites in the GP, LR, HR, and GX of Hongyang kiwifruit. The results suggested that metabolites in the four sub-tissues had different accumulation trends, and a large proportion of metabolites had high accumulation in the GP (Figure 1C). Therefore, we performed further analysis of metabolites that differentially accumulated in different sub-tissues.

### 3.2. Differentially Accumulated Metabolites Analysis

Univariate and multivariate statistical analyses were used to screen DAMs for each comparison group. The OPLS-DA and 200-response sorting test results showed that the model was stable and reliable (Figure 2A,B and Appendix A), and the VIP analysis in the OPLS-DA model could be used to screen DAMs. A total of 720 DAMs were screened from the 6 comparison groups under the set threshold. The DAMs in each comparison group are presented in Table 2. K-means analysis showed that the 720 DAMs could be divided into 6 accumulation patterns (Figure 3, Appendix A). Compared with the GP, LR, and HR, a total of 136 metabolites were highly accumulated in the GX, including 41 amino acids and derivatives, 16 flavonoids, 12 lignans, 10 alkaloids, 10 saccharides and alcohols, and 8 organic acids. Compared with the GX, LR, and HR, a total of 302 metabolites were highly accumulated in the GP, including 65 flavonoids, 62 phenolic acids, 62 terpenoids, 14 free fatty acids, 11 saccharides and alcohols, 9 organic acids, and 6 proanthocyanidins (Appendix A). These results indicated that the peel tissue of kiwifruit has potential value for development and utilization.

To further explore the potential mechanism of coloration and fruit quality formation of Hongyang kiwifruit, we specifically analyzed the differential metabolites in the LR and HR, which are the main edible tissue. A total of 251 DAMs were identified in the LR vs. HR comparison group, among them 211 DAMs were up-regulated in the HR, and only 40 DAMs were up-regulated in the LR (Figure 2C, Appendix A). KEGG enrichment analysis showed that these DAMs were mainly enriched in the phenylpropanoid biosynthesis, flavonoid biosynthesis, flavone and flavonol biosynthesis pathways (Figure 2D), and a total of 51 DAMs were obtained from KEGG pathway functional annotation information (Appendix A). The hierarchical clustering heatmap analysis showed that 43 metabolites out of 51 were highly accumulated in HR (Figure 4). For DAMs up-regulated in the HR, including 69 flavonoids, 53 phenolic acids, 38 terpenoids, key quality related metabolites (Tartaric acid, shikimic acid, ribitol), and key flesh color related metabolites (Cyanidin-3,5-O-diglucoside, quercetin-3-O-glucoside, kaempferol-3-O-glucoside, Pelargonidin-3-O-rutinoside-5-O-glucoside, Pelargonidin-3-O-glucoside). The DAMs up-regulated in the LR included 7 flavonoids, 6 terpenoids, 4 phenolic acids, and L-malic acid. Notably, there were no significant differences in the content of citric acid, quinic acid, glucose, fructose, and sucrose between the LR and HR (Figure 4, Appendix A).

### 3.3. Transcriptome Sequencing Analysis

A total of 12 cDNA libraries for the GP, GX, LR, and HR of Hongyang kiwifruit were constructed and sequenced to obtain gene expression signatures at the transcription level. The basis statistics of RNA-seq data are found in Appendix A. The average Q30 and GC content of the 12 libraries were 93.52% and 46.23%, respectively. A total of 48,139,469 clean reads were obtained and assembled into 45,659 unigenes, of which 42,097 (92.20%) unigenes were annotated (Appendix A). Among them, 41,986 (91.96%), 41,459 (90.81%), 35,577 (77.92%), 31,560 (69.12%), and 31,404 (68.78%) unigenes were annotated in the Nr, KOG, GO, Swissprot, and KEGG databases, respectively (Figure 5A). A total of 10,641 novel unigenes were identified in the RNA-seq data, and 6325 novel genes were annotated, which were mainly enriched in amino sugar and nucleotide sugar metabolism, oxidative phosphorylation, and starch and sucrose metabolism pathways (Figure 5B, Appendix A). In addition, a total of 2816 transcription factors were identified, mainly including MYB, bHLH, NAC, C2H2, and WRKY (Figure 5C, Appendix A).

### 3.4. Differentially Expressed Genes Analysis

The hierarchical clustering heatmap analysis showed that a large number of unigenes were highly expressed in the GP (Figure 6A), similar to the accumulation pattern of metabolites (Figure 1C). Differential expression analysis showed that a total of 8067 DEGs were identified in 6 comparison groups, with the threshold values of fold change ≥2 and *p*-value ≤ 0.05 (Appendix A). A large proportion of DEGs were obtained from the GP vs. GX, GP vs. HR, and GP vs. LR comparison groups, and only 695 DEGs and 605 DEGs were identified in the GX vs. HR and LR vs. HR comparison groups, respectively (Table 2). K-means analysis showed that the 8067 DEGs could be divided into 6 expression patterns, with 4121 DEGs highly expressed in the GP, and 1659 DEGs highly expressed in the GX (Figure 6B, Appendix A).

The hierarchical clustering heatmap of DEGs identified in the LR vs. HR comparison group showed that TLR 1, TLR 2, TLR 3 and THR 1, THR 2, THR 3 were divided into the same cluster, respectively, indicating that the identification results of DEGs were reliable (Figure 7A). By comparing the DAMs and DEGs obtained from the LR vs. HR comparison group, we observed that many DAMs and DEGs were enriched in the same KEGG pathway. Examples include flavone and flavonol biosynthesis, and hormone signal transduction pathways (Figure 7B). The results of correlation analysis also showed that a total of 510 DEGs and 85 DAMs were associated in the LR vs. HR comparison group (Figure 7C, Appendix A). For example, L-malic acid was associated with 121 DEGs (62 negatively associated, 59 positively associated) and tartaric acid was associated with 217 DEGs (135 negatively associated, 82 positively associated). quercetin-3-O-glucoside, kaempferol-3-O-glucoside, cyanidin-3,5-O-diglucoside, pelargonidin-3-O-glucoside, pelargonidin-3-O-rutinoside-5-O-glucoside, and cyanidin-3-O-rutinoside were associated with 31, 142, 154, 151, 96, and 47 DEGs, respectively. The DEG denoted as Actinidia18275 and annotated as 4-coumarate-CoA ligase [EC:6.2.1.12] was associated with MWSmce024, pmb0751, Lmmn001643, pmb3074, mws0906, and other 27 metabolites. Actinidia33689 annotated as cinnamyl-alcohol dehydrogenase [EC:1.1.1.195] was significantly associated 23 metabolites (Figure 7D, Appendix A). These results indicate that a complex regulatory mechanism exists in the variation of metabolites accumulated and gene expression abundance in Hongyang kiwifruit.

### 3.5. Integrated Analysis of the Transcriptome and Metabolome

To further identify candidate genes associated with flesh color and fruit quality, WGCNA was used to analyze the relationships between DEGs (obtained from the LR vs. HR comparison group) and the flavonol, flavone, flavanone, anthocyanin, soluble sugar, organic acid, and amino acid content for each sample. Eight co-expression modules were identified by WGCNA based on similar expression patterns (Figure 8A, Appendix A). The heatmap of module-trait correlations showed that the green, yellow, black, and red modules were highly positively correlated with the flesh color and fruit quality related metabolites, including eudesmol, costic acid, curcumenol, nootkation, salicylic acid, arbutin, quinic acid, pelargonidin-3-O-glycoside, quercetin-3-O-rutinoside, and kaempferol-3-O-rutinoside (Figure 8B). Combined with the correlation analysis of DEGs and DAMs, functional annotation results, and literature reports, a total of 23 key candidate genes associated with flesh color and fruit quality were screened (Appendix A). Seven transcription factors and one non-specific lipid-transfer protein genes were significantly up-regulated in the HR. Among them, *bHLH*, *WD40*, and *MYB5b* were all positively associated with cyanidin-3-O-rutinoside, *MADS*, *DELLA*, and *bZIP* were positively associated with cyanidin-3-O-rutinoside, kaempferol-3-O-glucoside, and tartaric acid, and negatively associated with L-malic acid. The transcription factors of *CYP82D47/78A9*, *ERF071/4*, and *WRKY52/48* were significantly up-regulated in LR, and significantly associated with malic acid and tartaric acid, ribitol and morin-3-O-arabinoside, and pelargonidin-3-O-glucoside (Appendix A).

### 3.6. qRT-PCR Verification

To further verify the reliability of candidate gene expression levels and transcriptome data, 12 representative candidate genes involved in flesh color and fruit quality were selected for qRT-PCR analysis. The results showed that the 12 candidate genes were all significantly differentially expressed in the LR and HR, and there were no significant differences in the expression of *ERF109*, *MYB5b* and *WD-40* between the GP and GX. In addition, compared with the LR, *CYP82D47*, *ERF109*, *WRKY52*, *CYP78A9*, *GATA9*, and *4CL* were significantly down-regulated in the HR, and *MADS*, *bHLH052*, *MYB5b*, *WD-40*, *bZIP44*, and *DELLA* were significantly up-regulated in the HR, which was consistent with the results of transcriptome analysis (Figure 9, Appendix A). Therefore, our RNA-seq data and screened candidate genes could be used for further regulatory mechanism analysis.

### 3.7. Spatial Metabolomic Analysis

To verify the accuracy of metabolite accumulation levels and tissue variability provided by the widely targeted metabolome, and to analyze the spatial distribution and dynamic changes during ripening in Hongyang kiwifruit, MALDI-MSI was used to visualize the spatial distribution of 7 key metabolites at three ripening phases (Appendix A). The ions corresponding to tartaric acid (*m*/*z* 149.03) and shikimic acid (*m*/*z* 173.06) were detected with increasing intensity in phase II and decreasing intensity in phase III, and the intensity in the HR was higher than in the LR. However, the ion intensity of citric acid (*m*/*z* 191.02) and malic acid (*m*/*z* 133.01) continued to decrease during ripening, and the intensity in the LR was higher than the HR. In contrast to citric acid and malic acid, the ion intensity of glucose (*m*/*z* 179.05), sucrose (*m*/*z* 341.09), and sorbitol (*m*/*z* 181.07) continued to increase during ripening, and were evenly distributed throughout the fruit (Figure 10). The spatial distribution of these metabolites was consistent with the metabolome data, and provided intuitive evidence for the physiological mechanism of postharvest kiwifruit quality formation.

## 4. Discussion

Kiwifruit is not only highly nutritious, but also has potential medicinal value [50]. Revealing the key metabolites that contribute to these properties is an important link to further improve the fruit quality of kiwifruit. Metabolomics, a technique to identify and quantify endogenous small molecule metabolites, has been widely used to identify key metabolites in a variety of horticultural plants [15]. Because kiwifruit is an edible fruit that needs to be peeled, previous studies on metabolite identification of kiwifruit mainly focused on the flesh [2,3]. In this study, metabolome analysis was performed on the peel, yellow-flesh, red-flesh, and core of Hongyang kiwifruit. A total of 1001 metabolites were identified, a considerably higher number than reported for Donghong (410 metabolites) [10] and previous reports in Hongyang (515 metabolites) kiwifruit [3]. This may be attributed to the maturity of metabolomics technology and the abundance of metabolite databases. A total of 952 metabolites were identified in the GP, mainly including 171 flavonoids, 135 phenolic acids, 126 lipids, 89 terpenoids, and 66 alkaloids. Among them, 34 metabolites were specifically accumulated in the GP (Appendix A). Differential accumulation analysis showed that compared with the GX, LR, and HR, 302 metabolites were highly accumulated in the GP, indicating that the GP of Hongyang kiwifruit is rich in a variety of active substances, and has potential value for pharmaceutical development and utilization. 

During the postharvest ripening, the outer flesh of Hongyang kiwifruit changed from light yellow in phase I to yellow in phase III, and the color of the inner flesh continued to deepen (Figure 1A), with significant accumulation of flavor substances. In this study, we analyzed the metabolite composition and content differences of mixed samples of yellow-flesh and red-flesh tissues at three different ripening phases. A total of 211 DAMs were highly accumulated in the HR, mainly including 69 flavonoids, 53 phenolic acids, and 36 terpenoids. Forty DAMs were highly accumulated in the LR, including 7 flavonoids, 6 terpenoids, and 4 phenolic acids (Appendix A). This preliminary result indicates that red-flesh is more abundant in metabolites and has better flavor and nutritional value than yellow-flesh, and flavonoids, phenolic acids, and terpenoids were the main contributors to this difference. In the future, we will further verify this conclusion by comparing the differences in metabolite components between red-core kiwifruit (such as Donghong, Wanhong, and Hongshen 2) and yellow flesh kiwifruit (such as Jinyan, Jinguo, and Jintao) varieties. In addition, the cyanidin-3,5-O-diglucoside, pelargonidin-3-O-rutinoside-5-O-glucoside, cyanidin-3-O-rutinoside, pelargonidin-3-O-glucoside, quercetin-3-O-diglucoside, and kaempferol-3-O-glucoside were up-regulated in the HR, and kaempferol-3-O-sambubioside and apigenin-8-O-glucoside were up-regulated in the LR (Appendix A). This means that the differential accumulation of these metabolites may be the key factors affecting the coloration of Hongyang kiwifruit [9,11]. 

To further clarify the potential regulatory mechanisms of postharvest color and quality formation in Hongyang kiwifruit, RNA-seq analysis was performed on the GP, GX, LR, and HR. A total of 45,659 unigenes were assembled of which 92.20% genes were annotated (Appendix A), showing that the sequencing and assembly results were reliable. Among them, 2816 transcription factors were identified. Many transcription factor family members have been reported to be involved in the biosynthesis and regulation of flavonoids. For example, MYBA1/MYBA2 can activate anthocyanin biosynthesis [51], MYB5a/MYB5b can regulate the expression of phenylpyruvate-like structural genes [52,53], MYB10 can bind to the promoters of UDP-glucose flavonoid 3-O-glycosyltransferase (UFGT) and anthocyanidin synthase (ANS) genes to regulate anthocyanin biosynthesis in fruit [54]. MYB114 inhibited the expression of anthocyanin accumulation enzyme dihydroflavonol 4-reductase [55]. Furthermore, bHLH3 and bHLH64 can promote the synthesis of proanthocyanidins and UFGT gene expression, respectively [56,57]. WD40 can interact with bHLH and MYB to form the MYB-bHLH-WD40 (MBW) complex, which is a key regulator to activate anthocyanin accumulation [58]. ERF105 inhibited anthocyanin synthesis by activating MYB140 and competing with MYB114 for binding to bHLH3 [59]. DELLA can directly chelate MYBL2 and JAZ repressors, resulting in the release of bHLH/MYB subunits and subsequent formation of active MBW complexes [60]. 

In this study, a total of 8067 DEGs were identified by comparative transcriptome analysis, and a large proportion of DEGs were highly expressed in the GP, which was consistent with the accumulation pattern of metabolites (Appendix A). These results further clarified the differences in metabolite accumulation and regulation pattern between the GP, GX, LR and HR of Hongyang kiwifruit, and the feasibility of molecular breeding for flesh color. In addition, the key transcription factors *bHLH*, *MYB5b*, *WD40*, *MADS*, *bZIP* and *DELLA* that regulate flavonoid biosynthesis were up-regulated in the HR, and positively correlated with malic acid, tartaric acid and anthocyanin. Some transcription factors, such as *CYP82d47/78A9*, *ERF071/4*, and *WRKY52/48*, which are significantly associated with the accumulation of organic acid, saccharides, and flavonoids were also identified to be up-regulated in the LR (Appendix A). Taken together, these genes may be involved in the formation of color and fruit quality of kiwifruit during postharvest ripening.

In this study, we carried out spatial imaging analysis on kiwifruit for the first time, and successfully screened 9-aminoacridine with no interference to the key flavor substances of kiwifruit, such as tartaric acid, malic acid, shikimic acid, citric acid, glucose, sucrose, and sorbitol. This study provided technical experience for subsequent research on spatial metabolism of kiwifruit, and provided a visual basis for the formation of quality of postharvest kiwifruit. At the same time, it can be mutually verified with the results of UPLC-ESI-MS/MS to further clarify the accuracy of the detection results. However, due to the biological characteristics of kiwifruit, section thickness has a certain influence on the visualization results of this study. For example, in the phase II tissue section, at a thickness of 40 μm, the red-flesh district lost a large amount of target metabolites, due to the radiation network structure of the fruit, resulting in a weak ion intensity. Therefore, the preparation of frozen sections and mass spectrometry imaging technology for kiwifruit needs to be further optimized.

## 5. Conclusions

This study investigated the differences in metabolite components and content in different tissues of Hongyang kiwifruit and the characteristics of the content changes of key flavor substances in kiwifruit during the postharvest softening process. A total of 302 metabolites were highly accumulated in the GP, indicating that the GP of kiwifruit has potential value for development and utilization. Tartaric acid, shikimic acid, ribitol, Cyanidin-3,5-O-diglucoside, Pelargonidin-3-O-rutinoside-5-O-glucoside, quercetin-3-O-glucoside, kaempferol-3-O-glucoside, and pelargonidin-3-O-glucoside are the key metabolites responsible for the difference in flesh color and flavor in different tissues of Hongyang kiwifruit. Shikimic acid and tartaric acid were mainly distributed in the HR, and showed a change pattern of first increasing and then decreasing during postharvest softening. Citric acid and malic acid were distributed throughout the fruit and continued to decrease during ripening. Glucose, sucrose, and sorbitol are distributed throughout the fruit and continued to increase during ripening. *CYP82D47/78A9*, *ERF109*, *WRKY52*, *GATA9*, *4CL*, *MADS*, *bHLH052*, *MYB5b*, *WD-40*, *bZIP44*, and *DELLA* may be involved in the regulation of coloration and fruit quality formation in Hongyang kiwifruit. Our study provides a basis for revealing the potential mechanism of coloration and fruit quality formation in *Actinidia chinensis* cv. Hongyang.

## Figures and Tables

**Figure 1 foods-13-00233-f001:**
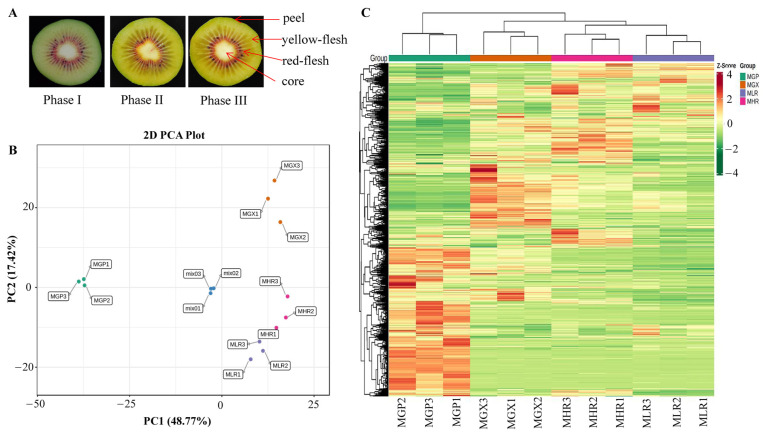
Cross-sectional plots of three ripening phases and sampling indicator plots of the GP, GX, LR, and HR in Hongyang kiwifruit (**A**). Principal component analysis (PCA) (**B**) and hierarchical clustering heatmap analysis (**C**) of metabolites identified in Hongyang kiwifruit. MGP, MGX, MLR, and MHR represent the metabolome data of these four sub-tissues, the same as follows.

**Figure 2 foods-13-00233-f002:**
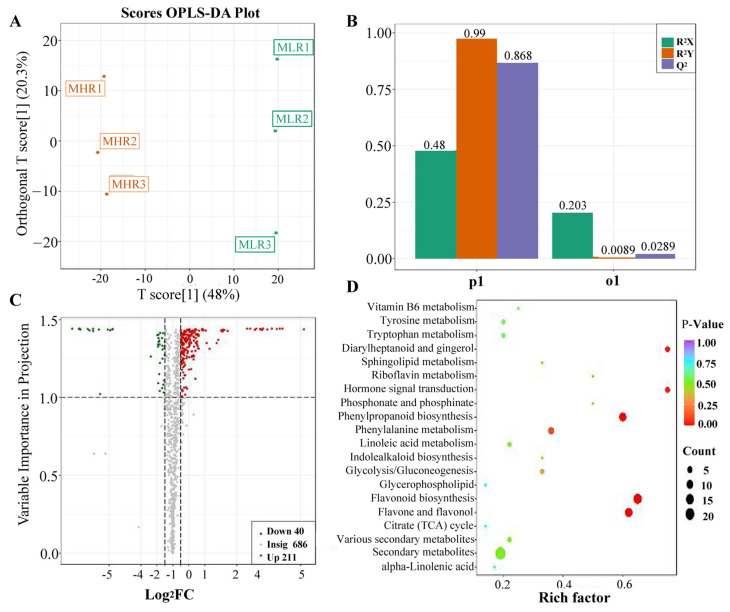
Analysis of differentially accumulated metabolites (DAMs) identified in the LR vs. HR comparison group. (**A**) Orthogonal partial least squares-discriminant analysis (OPLS-DA), (**B**) 200-response sorting tests, (**C**) volcano plots analysis, (**D**) KEGG enrichment analysis.

**Figure 3 foods-13-00233-f003:**
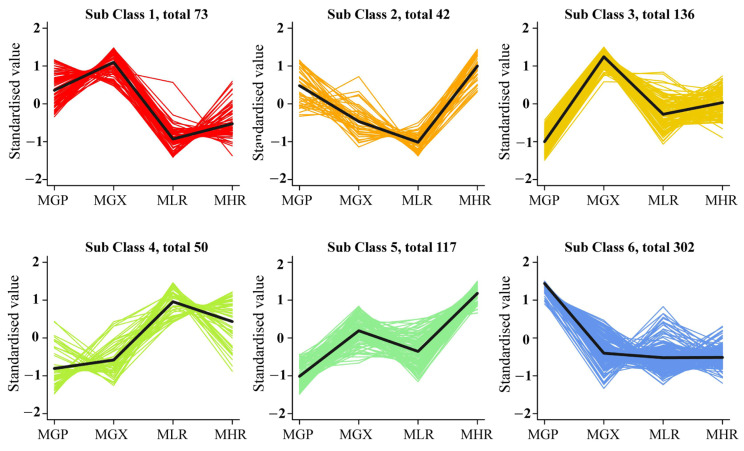
K-means analysis of 720 differentially accumulated metabolites (DAMs).

**Figure 4 foods-13-00233-f004:**
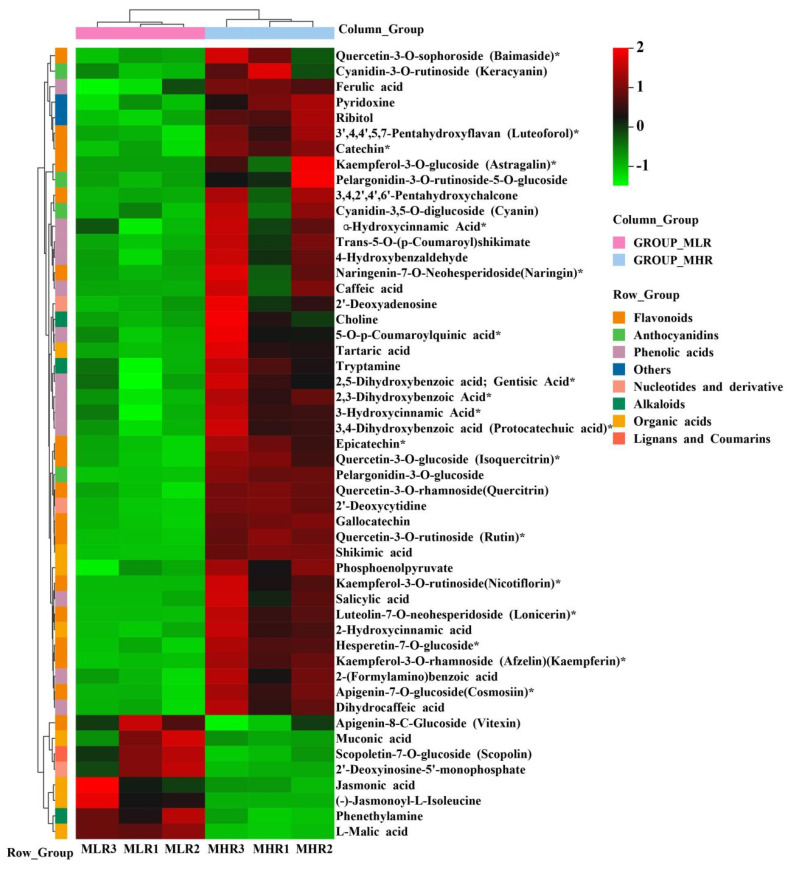
The hierarchical clustering heatmap analysis of 51 differentially accumulated metabolites (DAMs) obtained from KEGG pathway functional annotation. * indicates that the metabolite has an isomer.

**Figure 5 foods-13-00233-f005:**
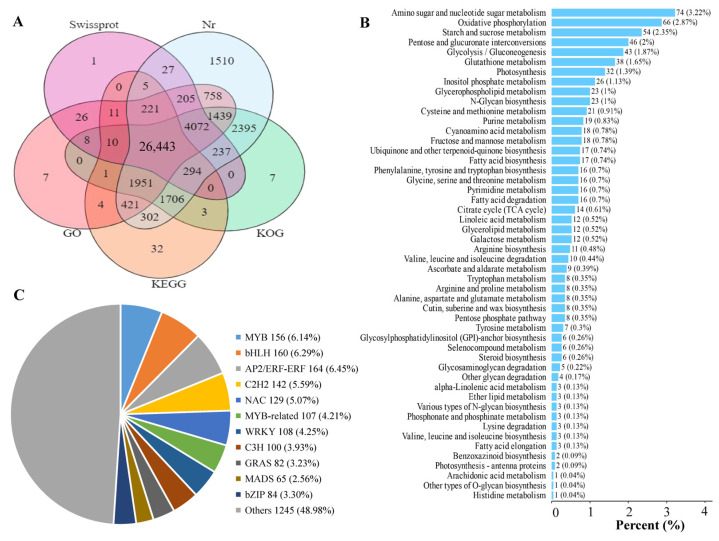
Statistical analysis of RNA-seq data. (**A**) Venn diagram analysis of the number of annotated genes in five public databases, (**B**) Enrichment analysis of novel genes, (**C**) Pie chart analysis of transcription factors.

**Figure 6 foods-13-00233-f006:**
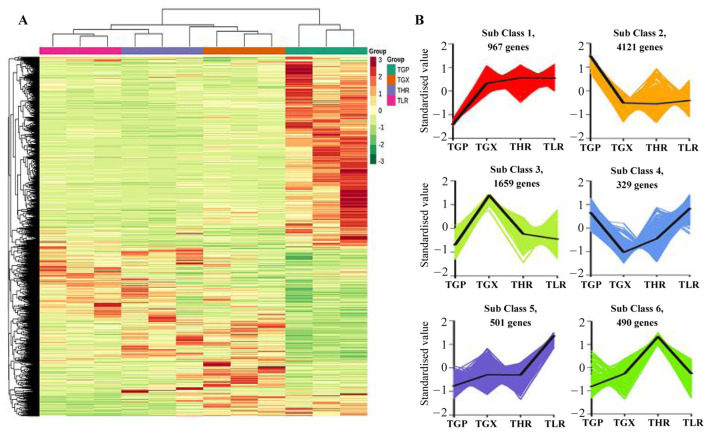
Hierarchical clustering heatmap analysis of assembled genes (**A**) and K-means analysis of differentially expressed genes (DEGs) (**B**). TGP, TGX, TLR, and THR represent the transcriptome data of these four sub-tissues, the same as follows.

**Figure 7 foods-13-00233-f007:**
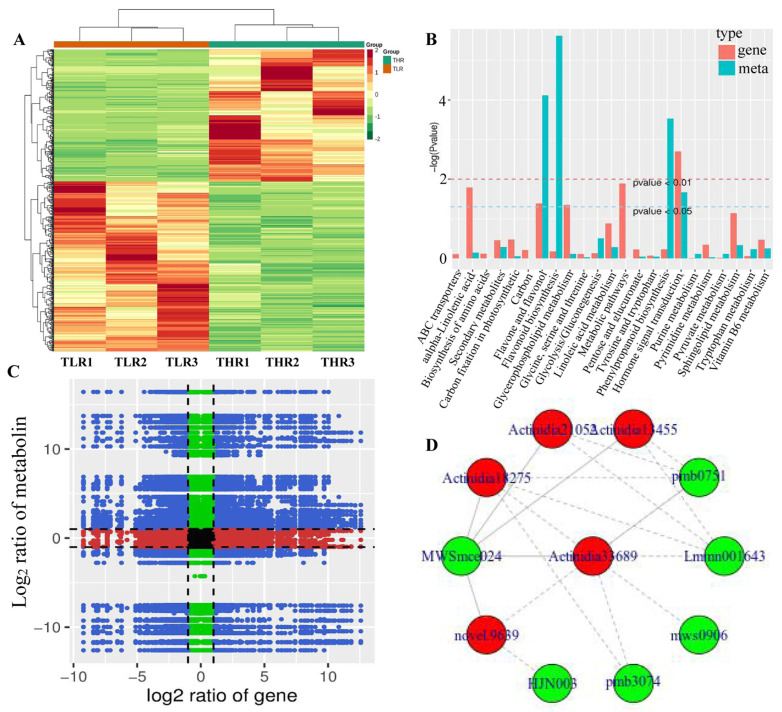
(**A**) Hierarchical clustering heatmap of differentially expressed genes (DEGs) identified in the LR vs. HR comparison group. (**B**) KEGG enrichment analysis of the DEGs (red column) and differentially accumulated metabolites (DAMs) (green column) that were enriched in the same pathway. (**C**) A nine quadrant diagram showing the correlation of DEGs and DAMs identified in the LR vs. HR comparison group. (**D**) Connection network between DEGs (red ovals) and DAMs (blue ovals).

**Figure 8 foods-13-00233-f008:**
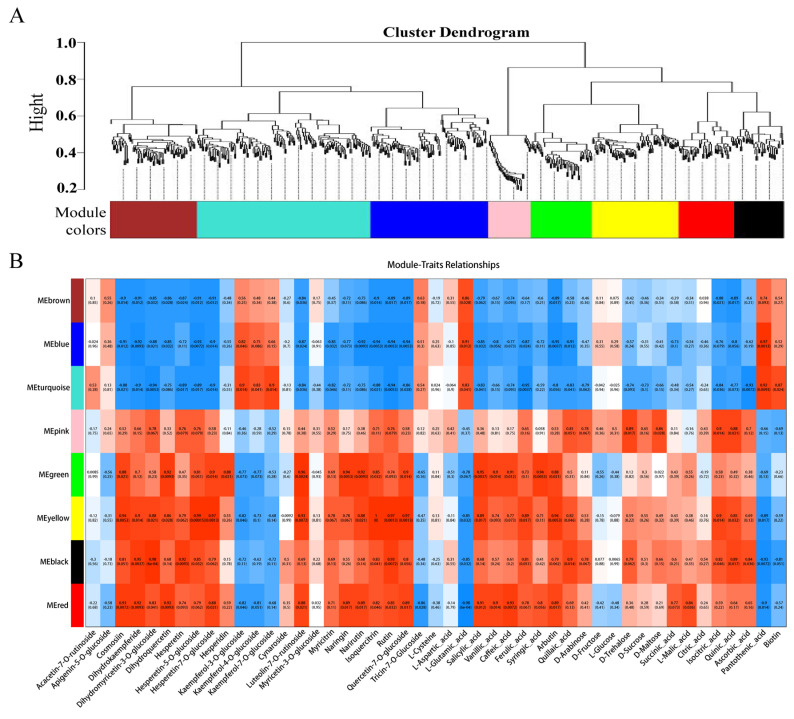
(**A**) The dendrogram of co-expression modules identified by weighted correlation network analysis (WGCNA), the major tree branches constitute 8 modules labeled with different colors. (**B**) The heatmap of module-trait correlations. Each column corresponds to a module indicated by different colors. Each row corresponds to a metabolite compound. Red and blue indicate positive and negative correlation, respectively.

**Figure 9 foods-13-00233-f009:**
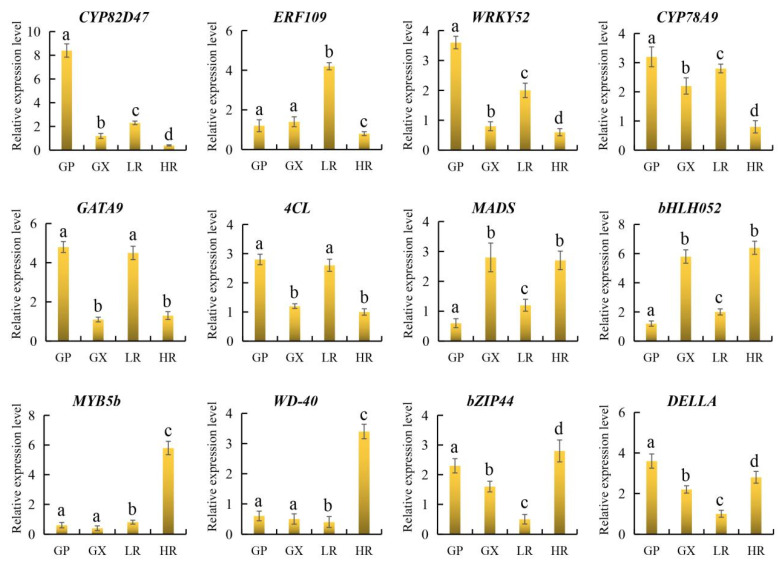
The relative expression level of 12 genes obtained by quantitative real-time PCR (qRT-PCR) analysis. The error bars represent the SD from three replicates. Lowercase letters represent the results of the difference significance analysis, with identical letters indicating insignificant differences and different letters indicating significant differences.

**Figure 10 foods-13-00233-f010:**
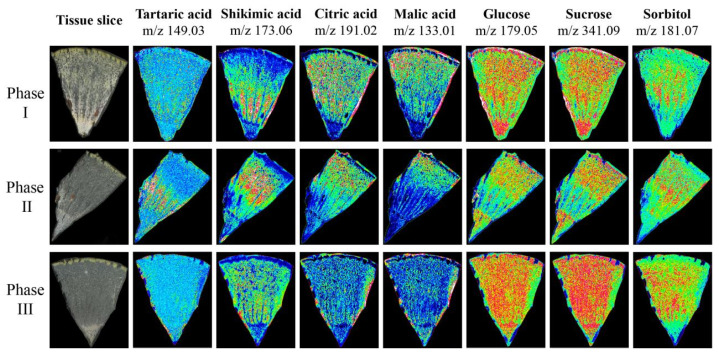
The mass spectrometry imaging of tartaric acid, shikimic acid, citric acid, malic acid, glucose, sucrose, and sorbitol in three phases. The blue (minimum) to green to red (maximum) scale indicates the distributed content of the target metabolite in the region.

**Table 1 foods-13-00233-t001:** Categories List of the 1001 Metabolites Identified in Four Sub-tissues of Hongyang Kiwifruit.

Category	Number	Category	Number
alkaloids	69	lignans and coumarins	55
amino acids and derivatives	86	Lipids	126
flavanols	26	nucleotides and derivatives	36
flavanones	19	organic acids	69
flavanonols	8	phenolic acids	137
flavones and Isoflavones	56	saccharides and alcohols	59
flavonoid carbonoside	9	Terpenoids	95
flavonols	62	Tannins	20
chalcones	10	Vitamin	10
anthocyanidins	12	Others	37

**Table 2 foods-13-00233-t002:** Summary of the Differentially Expressed Genes (DEGs) and Differentially Accumulated Metabolites (DAMs) Among Six Library Pairs.

Comparison Groups	Total	down_Meta	up_Meta	Total	down_Gene	up_Gene
GP vs. GX	491	261	230	4314	2821	1493
GP vs. HR	518	295	223	5123	3341	1782
GP vs. LR	506	343	163	4977	3275	1702
GX vs. HR	246	163	83	695	390	305
GX vs. LR	315	266	49	1819	989	830
LR vs. HR	251	40	211	605	340	265

## Data Availability

Data is contained within the article or Appendix A.

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
