# Peer review of "Combined Widely Targeted Metabolomic, Transcriptomic, and Spatial Metabolomic Analysis Reveals the Potential Mechanism of Coloration and Fruit Quality Formation in Actinidia chinensis cv. Hongyang"

_foods, 2024, doi:10.3390/foods13020233_

Round 1

Reviewer 1 Report

Comments and Suggestions for Authors

This study presents a comprehensive analysis of the Hongyang variety of kiwifruit, focusing on the differences in coloration and fruit quality between its outer yellow-flesh (LR) and inner red-flesh (HR) parts. Utilizing an array of methods including metabolomic, transcriptomic, and spatial metabolomic analyses, the research identifies 1,001 metabolites, with 211 having significantly higher accumulation in the HR. Key findings include the presence of flavonoids, phenolic acids, terpenoids, and anthocyanins in higher quantities. This research provides valuable insights into the mechanisms behind kiwifruit flesh color and fruit quality formation, offering a theoretical basis for future studies in this area. And I have some questions and suggestions for the authors, these are listed below:

The authors mentions that a total of 1,001 metabolites were identified in the Hongyang Kiwifruit, which is considerably higher than the number identified in other varieties like Donghong, how does this extensive variety of metabolites in Hongyang Kiwifruit, especially in the peel (GP), compare to other kiwifruit varieties in terms of contributing to the fruit's nutritional and medicinal properties?

The authors consider that the maturity of metabolomics technology and the expansion of metabolite databases have influenced these results?

The study discusses how the metabolite concentration and composition of Hongyang Kiwifruit's yellow- and red-flesh tissues vary as the fruit ripens through distinct stages. In specifically, how does the varied accumulation of phenolic acids, terpenoids, and flavonoids in the red and yellow flesh affect the fruit's overall flavor, taste, and nutritional value over the ripening phases?

The authors identify the particular metabolites and transcription factors involved in the postharvest ripening process and how it affects the color and quality of kiwifruit, but what do these results actually mean in terms of the food and agriculture sectors? How may this information be used to improve kiwifruit quality, shelf life, and storage in a commercial setting?

The article highlights the groundbreaking work done on kiwifruit spatial imaging analysis as well as the difficulties brought on by the biological traits of the fruit, such as how slice thickness affects visualization outcomes. What impact do these methodological and technical difficulties have on the precision and dependability of the spatial metabolism analysis in kiwifruit? What are some possible fixes or enhancements that might be implemented in subsequent studies to improve the accuracy of this analysis?

Author Response

Response to Reviewer 1 Comments

1. Summary

Thank you very much for taking the time to review this manuscript. Please find the detailed responses below and the corresponding revisions/corrections highlighted/in track changes in the revised manuscript.

2. Questions for General Evaluation

Reviewer’s Evaluation

Response and Revisions

Does the introduction provide sufficient background and include all relevant references?

Yes

Are all the cited references relevant to the research?

Yes

Is the research design appropriate?

Yes

Are the methods adequately described?

Yes

Are the results clearly presented?

Yes

Are the conclusions supported by the results?

Yes

 3. Point-by-point response to Comments and Suggestions for Authors

Comments 1: The authors mentions that a total of 1,001 metabolites were identified in the Hongyang Kiwifruit, which is considerably higher than the number identified in other varieties like Donghong, how does this extensive variety of metabolites in Hongyang Kiwifruit, especially in the peel (GP), compare to other kiwifruit varieties in terms of contributing to the fruit's nutritional and medicinal properties?

Response 1: Thank you for pointing this out. In this study, a total of 1001 metabolites were identified, which was much higher than that previously reported in Donghong. Through data comparison and analysis, we found that more abundant flavonoids, phenolic acids, lipids, amino acids and terpenoids were the main reasons for this difference. Lipids and amino acids are the basic components of fruit nutrition. Flavonoids, phenolic acids and terpenoids, as the most important bioactive substances in plant tissues, have anti-bacterial, anti-inflammatory, antiviral and anti-oxidation and other medicinal properties (we have added relevant statement in the introduction of revised manuscript). In addition, it plays an important role in the formation of pulp color, fruit quality and fruit aroma. In this study, we first measured and analyzed metabolites in kiwifruit peel (GP), and found that a large number of flavonoids, phenolic acids and terpenoids were enriched in GP. This indicates that the peel (GP) has great potential value for medicinal exploitation. In general, the rich variety and content of flavonoids, lipids, amino acids, phenolic acids and terpenoids contribute the fruit’s nutritional and medicinal properties. 

Comments 2: The authors consider that the maturity of metabolomics technology and the expansion of metabolite databases have influenced these results?

Response 2: Thank you for pointing this out. In our opinion, the previous detection of metabolites was usually performed using HPLC. In recent years, the use of UPLC-ESI-MS/MS has undoubtedly improved the detection range and accuracy. The qualitative and quantitative analyses of the metabolites were performed using secondary spectral information based on the public metabolite database and the self-built MWBD database. In recent years, a large number of metabolomics data of plant tissues have been published, which undoubtedly enriches the plant tissue metabolite database and improves the detection abundance of metabolite numbers. So we consider that the maturity of metabolomics technology and the expansion of metabolite databases have influenced these results regarding the number of metabolites.

Comments 3: The study discusses how the metabolite concentration and composition of Hongyang Kiwifruit's yellow- and red-flesh tissues vary as the fruit ripens through distinct stages. In specifically, how does the varied accumulation of phenolic acids, terpenoids, and flavonoids in the red and yellow flesh affect the fruit's overall flavor, taste, and nutritional value over the ripening phases?

Response 3: As we all know, kiwifruit is a kind of climacteric fruit. With the ripening of the fruit, the color of the flesh deepers, the sweetness of the fruit increases, and the fruit fragrance becomes richer. In addition to nutrients and flavor substances such as soluble sugars, organic acids, and amino acids, some flavonoids, phenolic acids, and terpenoids are also involved in this process. Such as, naringenin chalcone, esculeoside A/B for flavor formation, cyanidin 3-O-glucoside, pelargonidin 3-O-glucoside, quercetin rutinoside, kaempferol 3-O-glucoside for color formation, coumaric acid, coutaric acid, vanillin, and syringic acid for fruits taste, limonene, β-cyclocitral, linalool, benzaldehyde, and pulegone contribute to the fruits aroma. In short, some flavonoids, phenolic acids, and terpenoids are intrinsically important components of fruit flavor, taste, and nutritional.

Comments 4: The authors identify the particular metabolites and transcription factors involved in the postharvest ripening process and how it affects the color and quality of kiwifruit, but what do these results actually mean in terms of the food and agriculture sectors? How may this information be used to improve kiwifruit quality, shelf life, and storage in a commercial setting?

Response 4: To be honest, a lot of research on genes and their regulatory mechanisms is more about enriching the theoretical basis. In fact, it is very rare that target genes can be directly applied to food and agricultural production through genetic engineering technology. Of course, this is not to say that the study of functional genes does not have practical implications. The screening of key functional genes under specific conditions and the study of molecular mechanisms will help to guide the development of related technologies. Such as, in a recent study, we identified a key transcription factor that regulates fruit softening in Jinyan kiwifruit under suitable high temperature conditions. The downstream interacting proteins were identified by yeast two-hybrid assay, and it has been proved by relevant experiments that the protein plays an important role in the fruit softening process. Based on this finding, we identified two compounds FeSO4.7H2O and NaCl, which could significantly inhibit the protein activity and prolong the shelf life postharvest Jinyan kiwifruits (unpublished study). 

Comments 5: The article highlights the groundbreaking work done on kiwifruit spatial imaging analysis as well as the difficulties brought on by the biological traits of the fruit, such as how slice thickness affects visualization outcomes. What impact do these methodological and technical difficulties have on the precision and dependability of the spatial metabolism analysis in kiwifruit? What are some possible fixes or enhancements that might be implemented in subsequent studies to improve the accuracy of this analysis?

Response 5: The basic principle of MALDI mass spectrometry imaging is to mix the matrix molecules that can absorb ultraviolet laser and the sample to form co-crystals. The matrix molecules absorb the laser energy to assist the ion in the sample to ionize, and the mass spectrometer is used to separate and detect the test ion, and the mass spectrum associated with the spatial position of the sample is obtained. The spatial imaging map of the target metabolites on the tissue samples was obtained by software processing. Too thick slices can reduce the degree of ionization, which affects the detection accuracy of the target metabolites, as well as the imaging clarity. Due to the low density of many tissue samples, especially fruits, the sections are too thin, which will lead to the hollowness of the tissue sections in the subsequent experiments, thus the target metabolites cannot be detected. In terms of subsequent research and technology development, we believe that the detection accuracy of MALDI can be improved by optimizing the metabolite ionization method, upgrading the mass spectrometry imaging data analysis software, and improving the sectioning sample preparation technology. 

Reviewer 2 Report

Comments and Suggestions for Authors

Title : Combined Widely Targeted Metabolomic, Transcriptomic, and  Spatial Metabolomic Analysis Reveals the Potential Mechanism of Coloration and Fruit Quality Formation in Actinidia chinensis cv Hongyang.

This study aims to evaluate the relationship between the coloring of Actinidia chinensis cv Hongyang fruit and the existence of secondary metabolites. The results of this study showed that a total of 1,001 metabolites were identified in Hongyang kiwifruit fruit, of which 2011 metabolites were significantly higher in inner red-flesh than yellow-flesh, mainly include flavonoids, phenolic acids, terpenoids, and anthocyanins. This is an interesting study but requires clarification.

1.     This very important in the abstract to cite among the 211 metabolites in red fruit compared to yellow fruits, the most abundant metabolites in red flesh

2.     In the introduction, a paragraph on the nutritional and therapeutic importance of these red or yellow metabolites is very important for readers.

3.     Line 126; How were you sure that all the metabolites were extracted?

4.     Line 85; it is important to quantify this variation of these metabolites between the two fruit colors

Author Response

Response to Reviewer 2 Comments

1. Summary

Thank you very much for taking the time to review this manuscript. Please find the detailed responses below and the corresponding revisions/corrections highlighted/in track changes in the revised manuscript.

2. Questions for General Evaluation

Reviewer’s Evaluation

Response and Revisions

Does the introduction provide sufficient background and include all relevant references?

Can be improved

We have added relevant statements in the introduction of revised manuscript

Are all the cited references relevant to the research?

Yes

Is the research design appropriate?

Yes

Are the methods adequately described?

Yes

Are the results clearly presented?

Yes

Are the conclusions supported by the results?

Yes

3. Point-by-point response to Comments and Suggestions for Authors

Comments 1: This very important in the abstract to cite among the 211 metabolites in red fruit compared to yellow fruits, the most abundant metabolites in red flesh.

Response 1: Thank you for your comments and suggestions. among the 211 metabolites in red fruit compared to yellow fruits, the most abundant metabolites in red flesh have been cited in the abstract (line 19-20) by specific numbers. 

Comments 2: In the introduction, a paragraph on the nutritional and therapeutic importance of these red or yellow metabolites is very important for readers.

Response 2: Thank you for your comments and suggestions. The differential metabolites between red-flesh and yellow flesh of Hongyang kiwifruit were mainly flavonoids, phenolic acids, and terpenoids. A paragraph describing the nutritional and therapeutic of these metabolites were added to the introduction of the revised manuscript (Line 46-64). At the same time, relevant references have been cited and added to the reference list.

Comments 3: Line 126; How were you sure that all the metabolites were extracted?

Response 3: Thank you for pointing this out. Firstly, we believe that no extraction method can extract all the metabolites in plant tissues at the same time, because the molecular weight, polarity and morphology of different metabolites are very different. The methanol extraction method was used in this study. On the one hand, because of the strong molecular polarity of methanol, more metabolites can be extracted; on the other hand, most of the current studies on plant tissue metabolomics use methanol extraction (Wang et al., 2021, New Phytologist; Qiu et al., 2020, Horticulture Research; Li et al., 2021, Food Chemistry; Guo et al., 2020, Industrial Crops & Products; Park et al., 2016, BMC Genomics; Wang et al., 2019, Molecules; Qiu et al., 2020, Journal of Agricultural and Food Chemistry, and so on).

Comments 4: Line 85; it is important to quantify this variation of these metabolites between the two fruit colors

Response 4: Thank you for pointing this out. We also quite agree that it is important to quantify this variation of these metabolites between red-flesh and yellow-flesh of kiwifruit. The specific number of differential metabolites is described throughout the manuscript, such as on line 199, 229-235, and 261 of revised manuscript. However, due to the large number of differential metabolites, the fold change and corresponding P-value of each metabolite between red-flesh and yellow flesh were only listed in the Supporting information File 4.